# Repeat *Ascaris* challenge reduces worm intensity through gastric cellular reprograming

Yifan Wu[1], Charlie Suarez-Reyes[1], Nina L. Tang[1], Alexander R. Kneubehl[1,2], Jill E. Weatherhead[1,2,3,4]*

1 Department of Pediatrics, Division of Tropical Medicine, Baylor College of Medicine, Houston, Texas, United States of America, 2 William T. Shearer Center for Human Immunobiology, Baylor College of Medicine and Texas Children's Hospital, Houston, Texas, United States of America, 3 Department of Medicine, Section of Infectious Diseases, Baylor College of Medicine, Houston, Texas, United States of America, 4 National School of Tropical Medicine, Baylor College of Medicine, Houston, Texas, United States of America

* weatherh@bcm.edu

## Abstract

Ascariasis (roundworm) is the most prevalent parasitic nematode infection worldwide, impacting approximately 500 million people predominantly in low- and middle-income countries (LMICs). While people of all ages are infected with *Ascaris*, infection intensity (defined by worm burden) paradoxically peaks in pre-school and school-aged children but then declines with age. The cause of age-dependent *Ascaris* worm intensity is not well understood but may be dependent on cellular changes in mucosal barrier sites. We have previously found that the gastric mucosa is a critical barrier site for *Ascaris* infection as ingested *Ascaris* larvae use acidic mammalian chitinase (AMCase) secreted by gastric chief cells and acid secreted by gastric parietal cells to hatch. After hatching, larvae translocate across the gastric mucosa to initiate the larval migratory cycle. However, mucosal injury induced by administration of Tamoxifen results in cellular changes that impair *Ascaris* hatching and reduce larval translocation across the gastric mucosa. Since individuals in endemic settings often experience recurrent infection throughout their lives, we set out to determine how repeated *Ascaris* exposures affect the gastric mucosa and the intensity of resultant infections. In this study, we established a repeated *Ascaris suum* challenge mouse model and found that repeated *Ascaris* challenge caused cellular changes in the gastric mucosa which reduced worm intensity in the liver. Importantly, these decreases in infection intensity following repeated infections occurred independent of the adaptive immune response. These findings indicate that gastric cellular changes may be a key mechanism leading to the observed age-dependent *Ascaris* worm intensity changes from childhood to adulthood.

**Data availability statement:** All data are available in the figures, tables, and supplementary material. Sequencing data and RNAseq analysis is publicly available on NCBI through BioProject PRJNA1231119. Analytical scripts used in the analysis of the sequencing data are publicly available on GitHub at https://github.com/kneubehl/repeated-ascaris-infection-gastric-rnaseq-analysis.

**Funding:** Funding for the manuscript was provided by the National Institutes of Health National Institute of Allergy and Infectious Diseases K08AI43968 and the Pediatric Infectious Disease Society Foundation Pichichero Family Foundation Vaccines for Children Initiative Research Award to JEW. ARK was supported through the Infection and Immunity T32 Fellowship T32AI055413 at Baylor College of Medicine. The funders had no role in study design, data collection and analysis, decision to publish, or preparation of the manuscript.

**Competing interests:** The authors have declared that no competing interests exist.

## Author summary

Ascariasis (roundworm) predominantly impacts people living in low- and middle-income countries and poses a significant global threat to human health across the age-spectrum. While both adults and children can be infected with *Ascaris* worms, children harbor higher worm burden which leads to end-organ disease including liver and lung pathology, malnutrition, and growth restriction. However, the reason for this age-related difference in parasite burden is unclear. Here, we found that repeated *Ascaris* infection in mice resulted in lower worm burden compared to single infection models, suggesting that with repeated exposure, the mice acquired protection against high worm burden. We also found that this protection was a result of cellular changes to the gastric mucosa following repeated infections and was not associated with immunologic learning. These findings highlight the importance of the gastric mucosa as a barrier site for parasite infection and provide a critical foundation for understanding the role of *Ascaris* infection in other gastric pathologies.

## Introduction

Ascariasis (roundworm) is the most prevalent parasitic nematode infection worldwide, impacting approximately 500 million people [1,2]. In endemic regions, particularly in low- and middle- income countries (LMICs), people are infected with either *Ascaris lumbricoides* or *Ascaris suum* by oral ingestion of eggs from contaminated soil or water [3–6]. After ingestion, *Ascaris* larvae hatch in the gastrointestinal tract and initiate a transient, highly immunogenic, larval migration cycle through the host's liver, lungs, and eventually the intestines to develop into adult worms where they live for up to 2 years [7]. In endemic regions, individuals experience recurrent infection throughout their lives [8]. While people of all ages are infected with *Ascaris*, infection intensity (defined by worm burden) paradoxically peaks in pre-school and school-aged children but subsequently declines with age.

The cause of age-dependent *Ascaris* worm intensity is not well understood but likely results from a combination of changes in host behavior alongside changes to the adaptive immune response and/or cellular changes at mucosal barrier sites following repetitive infection. Since *Ascaris* is transmitted by oral ingestion of *Ascaris* eggs found in the environment, childhood behaviors such as pica (also known as geophagia) put younger children at risk of high worm intensity [9]. Meanwhile, adaptive immunity such as activation of CD4+ T cells [10] as well as IgE isotype class switching [11] may provide protective immunity against *Ascaris* at different immune barrier sites following recurrent infection. In a mouse model of *Ascaris* larval migration, recurrent infection was associated with reduced worm intensity in the lungs and was associated with an increased number of T cells, in addition to innate immune cells such as neutrophils, eosinophils and macrophages [10]. While each of these mechanisms are likely contributing to age-dependent differences in *Ascaris* worm intensity, evaluation of cellular changes at mucosal barrier sites has largely been unexplored.

The gastric mucosa is a critical barrier site for *Ascaris* infection. We have previously shown that following oral ingestion of *Ascaris* eggs, larvae use acidic mammalian chitinase (AMCase) secreted by gastric chief cells and acid secreted by gastric parietal cells to hatch [12]. Therefore, this hatching occurs in the host stomach despite prior literature suggesting larvae hatch in the intestines [12,13]. Once hatched, larvae translocate across the gastric mucosa to initiate the larval migratory cycle. Importantly, when we induced mucosal damage through administration of Tamoxifen, changes to the gastric mucosa resulted in reduced AMCase and gastric acid, impairing *Ascaris* hatching and reducing larval translocation across the gastric mucosa [14]. This resulted in decreased worm intensity in the liver and lungs during the larval migration cycle suggesting a critical role for the gastric mucosa in determining worm intensity [12]. The aim of this study was therefore to assess the mechanism by which gastric mucosa adaptations might contribute to age-dependent changes in worm intensity. Towards this goal, we established a repeated *Ascaris suum* challenge mouse model and evaluated its impact on the gastric mucosal barrier. We found that repeated *Ascaris* challenge caused cellular changes in the gastric mucosa which reduced worm intensity in the liver independent of the adaptive immune response.

## Results

### Repeat *Ascaris* challenge reduces *Ascaris* worm intensity in the liver

Prior literature has shown that *Ascaris* worm intensity in mice is reduced in the lungs following repeat *Ascaris* challenge [10]. However, the liver is the first site in the migration cycle. Therefore, we first asked if repeat *Ascaris* challenge in mice impacted larval intensity in the liver. Towards this, we used a repeat *Ascaris* challenge model in which wild-type mice were infected with 2500 embryonated *Ascaris* eggs twice a week for two weeks (a total of 4 challenges) by oral gavage (Fig 1A) compared to a single infection with *Ascaris* eggs [15]. Four days following the last *Ascaris* egg challenge, mice were euthanized. We found that mice exposed to repeat *Ascaris* challenge had reduced *Ascaris* larval burden in the liver compared to mice exposed to a single *Ascaris* challenge (Fig 1B).

Our prior work demonstrated that impairing gastric AMCase and acid secretion inhibited *Ascaris* hatching, larval translocation across the gastric mucosa, and migration to the liver and lungs [12]. Based on these data, we hypothesized that the stomach may be a primary barrier site involved in modulating *Ascaris* infection intensity through mucosal secretion of gastric acid and AMCase. To this end, we measured *atp4a,* a gene that encodes the proton pump responsible for gastric acid secretion*,* and *chia1*, a gene that encodes AMCase, mRNA expression from stomach tissue harvested from the wild-type repeat *Ascaris* challenge mouse model compared to wild-type, naive mice 4 days following the last *Ascaris* egg challenge. Indeed, *atp4a* and *chia1* mRNA expression levels measured by qPCR were both reduced in the repeat *Ascaris* challenge mice (Fig 1C), and decreased Atp4a and AMCase protein levels were confirmed by western blot (Fig 1D). Together these results demonstrate a reduction in *Ascaris* worm intensity in the liver following repeat *Ascaris* challenge that is associated with decreased gastric mucosa function, specifically reduction in gastric acid and AMCase secretion.

### Gastric CD4+ T cells are decreased following repeat *Ascaris* challenge

Following our findings that *atp4a* and *chia1* were reduced in repeat *Ascaris* exposed mice, we next aimed to evaluate changes in the gastric transcriptome following repeat *Ascaris* challenge. Using our wild-type repeat *Ascaris* challenge model (Fig 1A), we compared bulk RNA sequencing from stomach tissue harvested 4 days following the last challenge with stomach tissue from naïve mice. A total of 19,166 genes were detected. Considering all genes, stomach tissue harvested from repeat *Ascaris* challenge mice had a distinct transcriptome by Euclidean distance matrix S1A Fig) and principal component analysis (S1B Fig) compared to naive controls. Of those genes, 751 were significantly downregulated and 695 were significantly upregulated in the stomach of wild-type repeat *Ascaris* challenge mice compared to wild-type naive mice as observed by volcano plot (S1C Fig). This was further depicted via gene ontology analysis on involved pathways (S1D and S1E Fig).

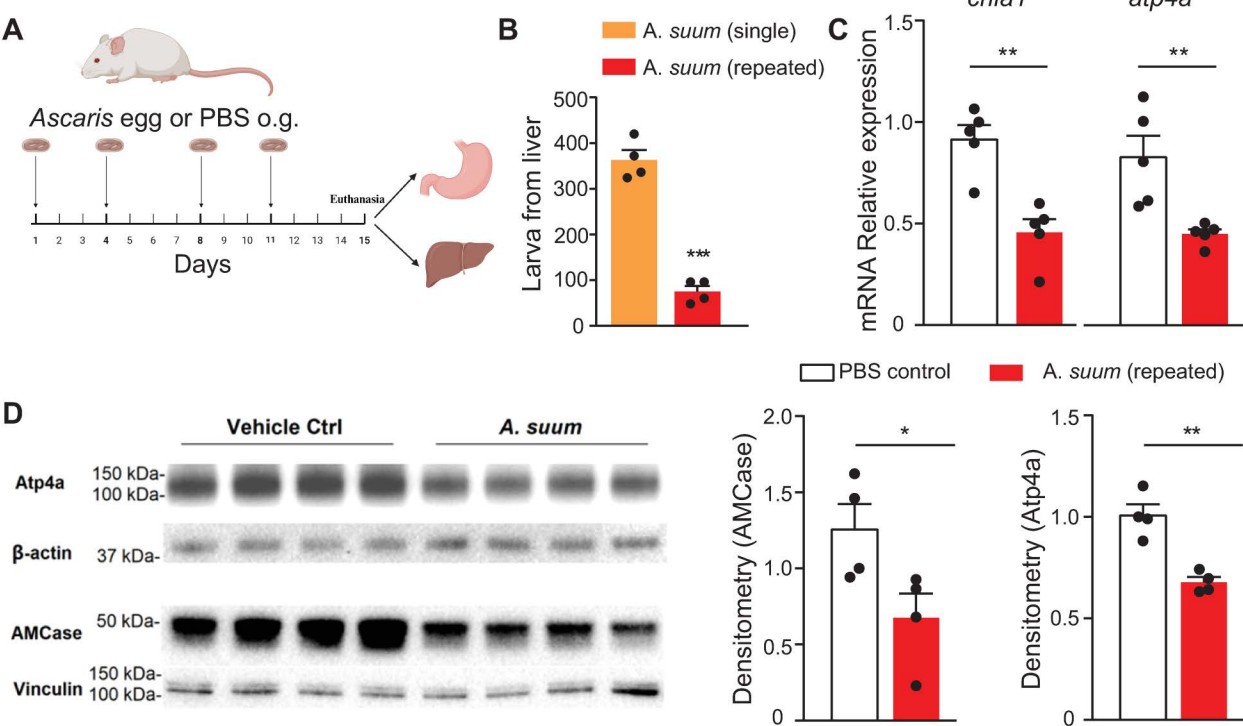

**Fig 1. Repeat *Ascaris* challenge by oral gavage reduces *Ascaris* worm intensity in the liver with associated impaired host gastric microenvironment function.** Repeat *Ascaris* infection mouse model (A) in which wild-type mice were infected with embryonated *Ascaris* eggs or PBS twice a week for two weeks by oral gavage (o.g.). Four days following the last infection mice were euthanized and stomach and liver tissue were harvested. Larval count in the liver (B) illustrating decreased intensity in the repeat *Ascaris* infection mouse model compared to a single *Ascaris* infection model. mRNA relative gene expression (C) of *chia 1* (left) and *atp4a* (right) normalized to 18s in the stomach by qPCR following repeated *Ascaris* infection compared to PBS, non-infected controls. Western blot with quantification (D) of protein expression of Atp4a and AMCase in the stomach following repeated *Ascaris* infection compared to PBS, non-infected controls. Relative expressions of both AMCase and Atp4a were calculated based on densitometry normalization to the first control sample. (n ≥ 4, mean±S.E.M, *p < 0.05, **p < 0.01, ***p < 0.001 using two-tailed Student's t-test. Data are shown as representative of two independent experiments. Illustration created by biorender.com.).

Based on the reported role of the adaptive immune response in controlling *Ascaris* infection [10], we hypothesized that recruitment and activation of CD4+ T cells would provide anti-*Ascaris* immunity in the stomach directly by killing the parasite or indirectly by altering gastric acid and AMCase secretion. To evaluate if gastric CD4+ T cells provide anti-*Ascaris* immunity following repeat *Ascaris* challenge, we identified T cell recruitment and activation genes in the stomach tissue from our RNA sequencing data. Surprisingly, genes involved in T cell recruitment and activation were downregulated in the stomach following repeat *Ascaris* challenge compared to naive controls (Fig 2A). The reduced expression of common T cell chemokine genes including *cxcl13, ccl21b, ccl5, cxcr3, ccr7, il16, xcl1* found in the transcriptomic analysis were confirmed by qPCR (Fig 2B). Given this result, we next asked if the lack of T cell recruitment and activation chemokine gene expression was indicative of reduced CD4+ T cells in stomach tissue. Using flow cytometry on stomach tissue from wild-type repeat *Ascaris* challenge mice compared to wild-type naive controls, we found that repeat *Ascaris* challenge was associated with reduced CD4+ T cells in stomach tissue (Fig 2C). These data suggest that gastric adaptive immune responses, specifically CD4+ T cells, do not play a role in anti-*Ascaris* immunity at the gastric mucosa.

To confirm that *Ascaris* larval hatching and translocation were not modulated by the adaptive immune response, particularly CD4+ T cells, we challenged mice deficient in T cells and B cells (RAG2-/- mice) with 2500 embryonated *Ascaris* eggs twice a week for two weeks (a total of 4 challenges) by oral gavage (Fig 2D). Despite deficiency in T cells and B

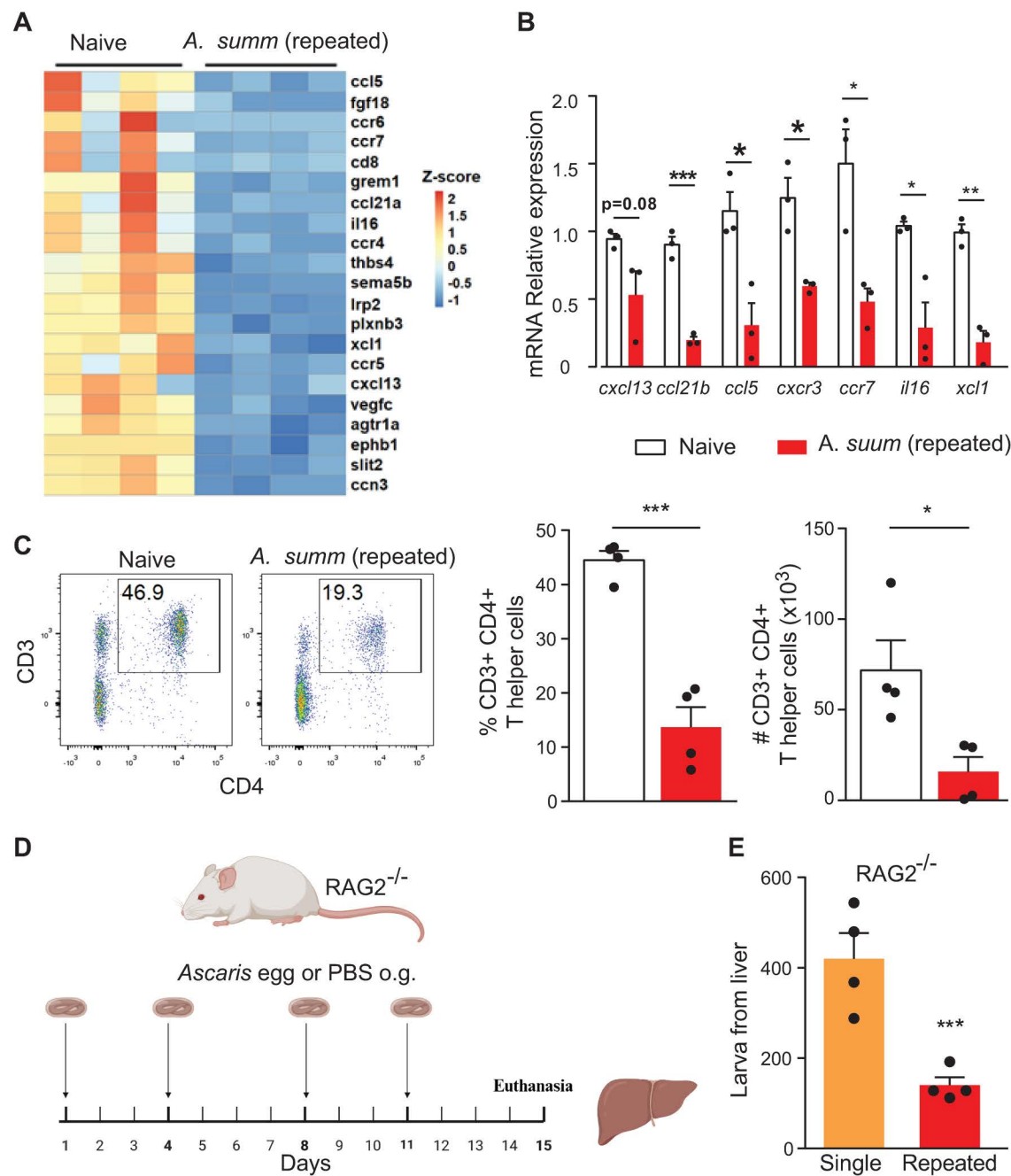

**Fig 2. Reduced worm intensity is independent of the gastric adaptive immune response following repeated *Ascaris* challenge.** Repeat *Ascaris* infection mouse model in which wild-type mice were infected with embryonated *Ascaris* eggs or PBS twice a week for two weeks by oral gavage. Four days following the last infection, mice were euthanized and stomach tissue was harvested. Repeat *Ascaris* infection is associated with reduced expression of T cell recruitment and activation genes in gastric tissue relative to infection-naïve mice represented by Z-scores (A) from bulk RNA sequencing. mRNA relative expression (B) of T cell recruitment and activation genes from A in gastric tissue by qPCR normalized to the first control sample. Flow cytometry with quantification (C) showing concentration of CD4⁺ T cells in gastric tissue. Repeat *Ascaris* infection mouse model (D) in which mice deficient in T and B cells (RAG2⁻/⁻) were infected with embryonated *Ascaris* eggs or PBS twice a week for two weeks by oral gavage. Four days following the last infection, mice were euthanized and liver tissue was harvested. Larval count in the liver of RAG2⁻/⁻ mice (E) illustrating decreased infection intensity for repeat *Ascaris* infection mouse model compared to a single *Ascaris* infection despite loss of adaptive immunity. (n ≥ 3, mean±S.E.M, *p < 0.05, **p < 0.01, ***p < 0.001 using two-tailed Student's t-test. Data are shown as representative of two independent experiments. Illustration created by biorender.com.).

cells, RAG2$^{-/-}$ repeat *Ascaris* challenge mice had reduced larval intensity in the liver compared to RAG2$^{-/-}$ mice exposed to a single *Ascaris* challenge (Fig 2E). Larval burdens in RAG2$^{-/-}$ repeat *Ascaris* challenge mice were similar to wild-type repeat *Ascaris* challenge mice (S1F Fig), confirming that gastric adaptive immune responses do not play an essential role in reducing larval worm intensity.

### Repeat *Ascaris* challenge causes gastric cellular reprogramming

Based on the evidence that the gastric adaptive immune response was not responsible for modulating *Ascaris* larval intensity (Fig 2E), the unique transcriptome observed in *Ascaris* infected versus uninfected controls and the decreased expression of *atp4a* and *chia1* evaluated with qPCR following repeated *Ascaris* challenge (Fig 1C), we hypothesized that the decreased larval intensity in our repeat challenge model was due to cellular changes in the gastric mucosal barrier. We therefore evaluated our stomach transcriptome data to assess genes associated with gastric cellular reprogramming and found decreased expression of parietal cell proton pump genes, *atp4a* and *atp4b* (Fig 3A), concordant with our prior qPCR data. Decreased parietal cell gene expression was accompanied by increased expression of genes involved in apoptosis by bulk RNA sequencing (Fig 3B). Given the combination of relatively increased apoptotic pathways and decreased parietal cell function in our repeat challenge model, we evaluated gastric histopathology from wild-type repeat *Ascaris* challenge mice compared to wild-type naive mice (Fig 3C). We found that repeat *Ascaris* challenge mice had increased apoptotic bodies in the gastric tissue consistent with parietal cell loss (Fig 3D).

Based on the decreases in *chia1* mRNA expression by qPCR in our repeat challenge model, we hypothesized that chief cells were undergoing transdifferentiation. Indeed, stomach transcriptomic data demonstrated increased gene expression of trefoil factor 2 (*tff2*) also known as spasmolytic polypeptide and *muc1* as well as decreased gene expression of *mist1* (Fig 3A) following repeat *Ascaris* challenge. Increased expression of *tff2* suggests gastric chief cells are transdifferentiating into spasmolytic polypeptide-expressing metaplasia (SPEM) which classically occurs following parietal cell loss. Upregulation of *tff2* was visualized in the gastric crypts by immunohistochemistry staining (Fig 3E) and quantitated by RNA expression using qPCR (Fig 3F). Together these data demonstrate that repeat *Ascaris* challenge induces cellular reprogramming of the gastric mucosa known as pyloric metaplasia, apoptosis of parietal cells and transdifferentiation of chief cells, suggesting a novel mechanism that leads to differences in *Ascaris* intensity.

## Discussion

We found that repeat *Ascaris* infection reduces worm intensity through the development of pyloric metaplasia, gastric cell reprogramming following mucosal injury that leads to parietal cell apoptosis and chief cell transdifferentiation into SPEM cells. These gastric cellular changes resulted in reduced gastric acid and AMCase secretion in the stomach which impaired *Ascaris* egg hatching and translocation across the gastric mucosa. This ultimately results in reduced larval intensity in the liver. Interestingly, these changes occurred independent of the adaptive immune response. While future studies are needed to compare the impact of uninfected, single-infection and chronic, repeated infection of *Ascaris* on gastric cellular changes, these findings highlight for the first time the role of the gastric mucosa as a primary barrier site in the prevention of heavy *Ascaris* infections following repeated exposures.

Our finding that repeated challenge with *Ascaris* induces the host gastric mucosa to undergo cellular changes which reduce AMCase and, thus, prevent *Ascaris* infection, raises the possibility that gastric cellular changes may be a conserved anti-nematode mechanism to reduce parasite burden in the host. Nematode eggs are covered in a thick chitin exterior that protects the egg in harsh environments but conversely requires a mechanism within the host to break down this chitin exterior to facilitate infection. We have previously shown for *Ascaris* infection that gastric AMCase serves this role, breaking down the chitinous egg [12]. Due to similar egg composition amongst different nematode species, it is reasonable to hypothesize that other nematodes may also use AMCase for hatching and that the gastric cellular changes observed here would confer host resistance to other nematodes in a similar mechanism. If true, this mechanism could be

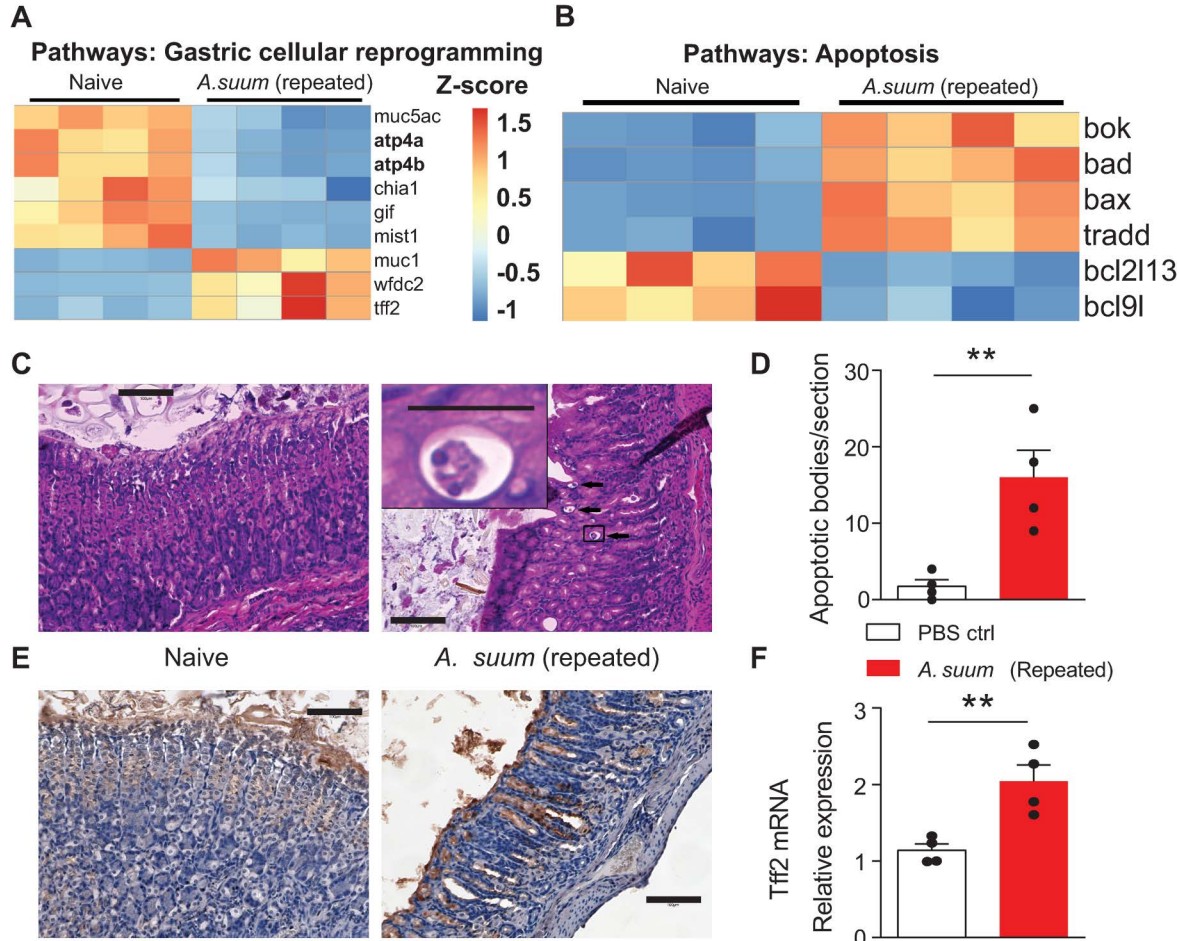

**Fig 3. Repeated *Ascaris* challenge causes cellular changes to the gastric mucosa.** Bulk RNA sequencing shows differential expression represented by z-score (A) for genes in stomach tissue associated with cellular reprogramming and development of pyloric metaplasia following mucosal injury including relatively decreased expression of atp4a and atp4b and increased expression of tff2 in the repeat *Ascaris* mouse models compared to naïve controls. Bulk RNA sequencing in stomach tissue also reveal (B) increased expression (represented by Z-score) of genes involved in cellular apoptosis pathways in repeat *Ascaris* infected mice compared to uninfected controls. Immunohistochemistry (C) shows increased apoptotic bodies (insert) consistent with parietal cell death identified by histopathology (H&E staining) in wild-type repeat *Ascaris* challenge model compared to wild-type naïve mice (Scale bar: 100μm for 100x, 25 μm for insert). Quantification of apoptotic bodies (D) in the gastric mucosa supports increased cell death in the gastric mucosa following repeat *Ascaris* challenge compared to naive mice. Immunohistochemistry (E) demonstrates increased expression of tff2 (brown staining, Scale bar: 100μm) and qPCR (F) quantification of increased tff2 mRNA relative expression normalized to 18s in the gastric mucosa in repeated *Ascaris* challenge model compared to naive mice. (n = 4, mean±S.E.M, **p < 0.01, using two-tailed Student's t-test. Magnification: 100× and 400× with 5×zoom in. Scale bar: 100μm and 25 μm. Data are shown as representative of two independent experiments.).

targeted for the development of a pan-nematode vaccine which has been hampered by the complexity of the nematode life cycle and the host systemic immune response mounted during infection [16–19].

Gastric mucosal injury triggers the development of gastric cell reprogramming known as pyloric metaplasia to create an environment that rapidly protects and restores the mucosal barrier. Interestingly, these cellular changes also lead to overexpression of surface receptors, such as selectins like sialyl-LewisX receptor and integrins, used by *Helicobacter pylori* to adhere to and colonize the gastric mucosa [20,21]. Adherence to the gastric corpus epithelium allows *H. pylori* colonization and expansion of pyloric metaplasia in the gastric corpus in a positive feedback loop [22,23]. The type of gastric cellular changes we observed following *Ascaris* infection would therefore suggest that individuals who have endured repeat *Ascaris* infection

and subsequent *Ascaris*-induced pyloric metaplasia would have a higher risk for *H. pylori* colonization. Indeed, the geographic niche of these pathogens overlaps and co-infection has been documented in the literature [24]. However, no studies to date have examined the mechanisms that influence *Ascaris* and *H.pylori* co-infection. Interestingly, compared to children living in high-income countries, those living in LMICs are more commonly infected with *H. pylori* in early childhood [25,26]. By adulthood, up to 50–80% of individuals in LMICs are colonized with *H. pylori* compared to <40% of individuals in high-income countries [27,28]. The reasoning behind the earlier colonization and more prevalent disease in LMICs remains unknown. However, *Ascaris*-induced pyloric metaplasia at an early age may influence early childhood colonization with *H. pylori* as well as the overall lifetime risk of *H. pylori* associated morbidity and mortality such as gastric cancer. This finding would have a significant impact on public health efforts in *Ascaris* and *H. pylori* endemic regions of the world.

Based on our findings, it is likely that pyloric metaplasia not only aids in gastric repair and restoration following mucosal injury but also serves as an anti-*Ascaris* mechanism to prevent high worm intensity following repeated *Ascaris* infection over time. Thus, in endemic regions, where individuals experience recurrent infection throughout their lives, pyloric metaplasia is a likely contributor to the observed age-dependent *Ascaris* worm intensity changes from childhood to adulthood. In this study, mice were infected over a two-week period. While this model recapitulates the real-world experiences of individuals spending shorter periods in *Ascaris* endemic settings such as travelers, military personnel, and public health workers, this acute re-infection model has limited relevance to long-term residents with chronic re-exposure. Instead, a chronic repeat infection model over months rather than weeks in mice of different ages will provide a more relevant evaluation of the long-term cellular changes that occur in the gastric mucosa of individuals with prolonged exposures in endemic regions. This short-term repeated infection model will also need to be compared to a single-infection model in mice of different ages to differentiate the impacts of single and repeated *Ascaris* infections on the gastric mucosa. Additionally, while *Ascaris*-induced pyloric metaplasia prevents high larval intensity following repeat *Ascaris* infection, pyloric metaplasia may also lead to collateral gastric pathology. Although there is evidence of *H. pylori* and *Ascaris* co-infection and there are high rates of gastric pathology in LMICs, to date, there is no human connection between these health outcomes. Further evaluation of the collateral damage to the gastric mucosa and the downstream sequalae to the gastrointestinal tract as a result of *Ascaris*-induced pyloric metaplasia is critical to understanding the complex relationship between *Ascaris* and the gastric mucosal barrier.

## Methods

### Ethics statement

All mice were housed at the American Association for Accreditation of Laboratory Animal Care-accredited vivarium at Baylor College of Medicine under specific-pathogen-free conditions. Upon arrival, complete randomization of mice into longitudinal groups was performed. All experimental protocols were approved by the Institutional Animal Care and Use Committee of Baylor College of Medicine and followed federal guidelines (AN-6297).

### Mice

Ten 8-week-old BALB/c female mice (wildtype or Rag2-/-) were purchased from Jackson Laboratories (cat: 000651) or Taconic Biosciences (cat: 601). Upon arrival, complete randomization of mice into longitudinal groups was performed. Only female mice were used to ensure consistency in infectious burden [29]. All mice were housed in a vivarium under specific-pathogen-free conditions. All experimental protocols were approved by the Institutional Animal Care and Use Committee of Baylor College of Medicine and followed federal guidelines.

### *A. suum* experimental murine model

*A. suum* eggs were obtained from adult female worms from infected pigs in the Weatherhead laboratory at Baylor College of Medicine. Briefly, adult female worms were isolated and dissected to remove the uterus. The uterus was then strained through a filter to release unembryonated eggs. The eggs were washed with PBS three times

and subsequently resuspended in sulfuric acid for 60 days to allow for embryonation. For single infections, BALB/c mice were treated with a single inoculum of 2,500 embryonated *A. suum* eggs via oral gavage or PBS as previously described [12,15]. For repeated infections, BALB/c mice were infected with an inoculum of 2,500 embryonated *A. suum* eggs or PBS via oral gavage twice per week for 2 weeks. The infectious dose of *Ascaris* has been standardized in the literature in order to replicate human disease in a murine model [15,30]. The *A. suum* life cycle in a murine model mimics the life cycle in humans and has been previously described [15,31]. Following oral gavage of *A. suum* eggs or PBS, mice were euthanized at 4 days post last infection (p.i.) and stomachs and liver were harvested in preparation for experiments described below. For stomach tissue, the forestomach was removed and discarded and the remaining tissue was washed in PBS.

## Quantitative PCR

Stomach tissue was homogenized in TRIzol (15596026, Thermofisher scientific, Waltham MA) using M tubes on gentle-MACS dissociator (130-093-236, Miltenyi, Bergisch Gladbach, Germany) for RNA extraction following the TRIzol RNA extraction manual. Relative expression of mRNA for *chia1* and *atp4a* to 18s was detected by two-step, real-time quantitative reverse transcription-polymerase chain reaction (RT-PCR) with the 7500 Real-Time PCR System (Applied Biosystems, Foster City, CA) using Taqman probe (Mm00458221, Mm00444417, Hs03003631 Invitrogen, Carlsbad, CA) and TaqMan Fast Advanced Master Mix for qPCR (4444557, Thermo Fisher Scientific, Waltham MA) [32,33]. Relative expression was calculated based on $\Delta\Delta ct$ to control group.

## Western blot

Stomach tissue was homogenized in tissue lysis buffer (50mM NaCl, 20mM HEPES, 1mM EDTA, 2% Triton-X 100, 10% glycerol, with proteinase and phosphatase inhibitor) for protein extraction [34].

Protein expression levels of AMCase and Atp4a were detected using western blot. (4%-12% Nupage Bis-Tris gel, Thermo Fisher Scientific, Waltham MA)(Antibody: ab207169, ab174293, 1:2000, Abcam, Cambridge MA; #4970S, #13901T, 1:2000 Cell signaling, Danvers MA). Secondary antibody was goat anti-rabbit IgG, HRP (1:10000, 31460, Thermo Fisher Scientific, Waltham MA). Band intensity was calculated based on ImageJ and normalized to control group.

## Histopathology

Gastric tissue was fixed in 10% neutral-buffered formalin solution, processed and embedded in paraffin. 5 µm sections were cut, and slides were stained with H&E. Apoptotic bodies were numerated from each section from each individual mouse. Alternatively, gastric tissue was processed and embedded in paraffin, 5 µm sections were cut and immunohistochemistry was completed. Briefly, slides were deparaffinized, then permeabilized using 0.2% Triton X 100 (X100, Sigma-Aldrich, St. Louis, MO). After that, antigen recovery was completed using Diva Decloaker (DV2004, Biocare Medical, Pacheco, CA), and peroxidase blocked using 3% hydrogen peroxide. Slides were then blocked using 5% bovine serum albumin (A0100-005, Gendepot, Baker, TX) for 1 hour at room temperature, and incubated with primary anti-Tff2 antibody in 5% bovine serum albumin (1:50, 13681–1-AP, Thermofisher scientific, Waltham MA) overnight at 4 °C. Slides were then incubated and stained using ABC-HRP and DAB kit (PK-6200, SK-4100, Vector Laboratories, Newark, CA). Slides were then counterstained with hematoxylin, dehydrated and mounted.

## Isolation of *A. suum* larvae from the liver

Mice infected with *A. suum* eggs were euthanized 4 days p.i. to assess larval burden in the liver. We measured larval burden in the liver at day 4 p.i. because this is when intensity peaks [15]. Liver tissue was harvested and macerated with scissors, suspended into pre-warmed PBS and transferred into a modified Baermann apparatus. The collection system was then filled up to 40 ml of pre-warmed PBS and incubated at 37°C for 4 hours. Following incubation, the solution containing

larvae was collected from the apparatus, centrifuged at 800 xg for 5 minutes at room temperature, and washed with water to remove red blood cells. Larvae were washed with PBS for 3 more times and counted under the microscope.

## Bulk RNA sequencing

Stomach tissues from mice were collected 18 days after the initial *A. suum* infection*,* which was 4 days after the last infection*,* into 1mL of 1x DNA/RNA Shield reagent (R1100-50, Zymo Research, Irvine, CA) in a 2mL ZR bashingbead lysis tube (S6012-50, Zymo Research, Irvine, CA). The tissues were lysed using a Precellys 24 homogenizer (03119.200. RD000, Bertin Technologies, Montigny-le-Bretonneux, France) using the 6m/sec setting for 40s. The homogenates were transferred to 1.5mL DNA LoBind tubes (022431021, Eppendorf, Hamburg, Germany) and shipped on dry ice to SeqCenter (Pittsburgh, PA) for RNA extraction, polyA purification, and RNA sequencing. RNA was extracted using the Quick-RNA miniprep kit (R1054, Zymo Research, Irvine, CA). PolyA selection and library preparation for RNA sequencing were performed using the Illumina stranded mRNA prep kit, 2x150bp reads (20040534, Illumina, San Diego, CA). RNA sequencing was performed on a NovaSeq X Plus instrument. Each sample was sequenced using SeqCenter's 50M paired end, polyA read option and on average 20 Gbp of read data were generated for each sample. Read data were submitted to the Sequence Read Archive (BioProject PRJNA1231119).

RNA sequencing analysis was performed using Nextflow's nf-core/rnaseq pipeline (v3.14.0) [35] to generate read counts using the STAR aligner [36] and salmon [37] for gene counts. Differential gene expression analysis was performed using DEseq2 [38]. Exploratory data analysis from DEseq2 (PCA and Euclidean distance matrix analysis) indicated two outlier samples, one in each group. The DEseq2 analysis was performed again with these outliers removed and all further analysis was done without the outliers. To visualize the differentially expressed genes, a volcano plot was generated using the EnhancedVolcano R package [39]. Significantly differentially expressed genes (|log2 fold change|>2 and <0.05 p adjusted value) were highlighted in figure. Gene ontology (GO) analysis was performed using clusterProfiler R package [40]. Heatmaps of gene expression for selected genes were generated using the pheatmap R package [41]. Analytical scripts along with version information and the R environment used can be found on GitHub (https://github.com/kneubehl/repeated-ascaris-infection-gastric-rnaseq-analysis). The DEseq2 normalized counts for each sample were submitted to the Gene Expression Omnibus repository and are publicly available (PRJNA1231119).

## Single cell suspension

Stomach tissues from mice post multiple infections were collected 4 days p.i. Stomachs were cut into small pieces and incubated in digestion buffer (2mg/ml collagenase (#LS004177, Worthington), 0.04mg/ml DNAse (#10104159001, Sigma) and 20% FBS in HBSS for 1h at 37°C after which they were disaggregated by pressing through a 40 µM nylon mesh and centrifuged at 400×g for 5 minutes at 4°C. Supernatants were discarded, and 1.5mL of ACK (Thermofisher scientific, Waltham MA) was added and incubated for 3min at room temperature for erythrocyte lysis. ACK was then neutralized with 7.5mL of complete RPMI-1640 (Corning, NY), with 10% FBS and 1% Pen Strep, Gibco, Waltham MA). The resulting single cell suspension was prepared for flow cytometry analysis [42].

## Flow cytometry

Total stomach cells isolated above were stained with Live/Dead Fix Blue (L34961, Thermofisher scientific, Waltham MA) and CD45 (103112, Biolegend, San Diego, CA). For T helper cell staining, cells were stained with CD3 and CD4 (100222, 100412, Biolegend, San Diego, CA). [42,43].

## Statistical analysis

Data are presented as means ± standard errors of the means. Significant differences relative to PBS-challenged mice are expressed by P values of <0.05, as measured two tailed Student's t-test, one-way or two-way ANOVA followed by Tukey's

test for multiple comparison. Data normality was confirmed using the Shapiro-Wilk test. Experiments were repeated at least twice. All data points in the manuscript identify biological replicates. Sample size was determined based on preliminary data and was powered to detect a minimum effect size of 1.79-fold with 80% power at a significance level (α) of 0.05. These parameters guided our choice of using n ≥ 4 group.

## Supporting information

**S1 Fig. Repeated *Ascaris* challenge is associated with a distinct gastric transcriptomic profile, and larval burden in the liver is independent of adaptive immunity.** Heat map (A) and principal component analyses (B) demonstrate stark variations in transcriptomes based on infection status. Volcano plot (C) shows significantly differentially expressed genes (|log2 fold change| > 2 and <0.05 p adjusted value) in the gastric mucosa of mice repeatedly challenged with *Ascaris* compared to naïve mice. Gene ontology (GO) analysis illustrate (D) significantly downregulated and (E) upregulated pathways in infected gastric tissue compared to naive tissue. (F) There was no difference in *Ascaris* larvae intensity in the liver between wild-type and RAG2$^{-/-}$ mice following single and repeat *Ascaris* infection. (n = 4, mean±S.E.M, n.s.: not significant, using two-way ANOVA. Data are shown as representative of two independent experiments.). (TIF)

## Acknowledgments

This manuscript was prepared with the assistance of a science writer, Ariel M Lyons-Warren.

## Author contributions

**Conceptualization:** Yifan Wu, Jill E. Weatherhead.

**Data curation:** Yifan Wu, Charlie Suarez-Reyes.

**Formal analysis:** Yifan Wu, Alexander R. Kneubehl, Jill E. Weatherhead.

**Funding acquisition:** Jill E. Weatherhead.

**Investigation:** Yifan Wu, Charlie Suarez-Reyes, Alexander R. Kneubehl.

**Methodology:** Yifan Wu, Alexander R. Kneubehl, Jill E. Weatherhead.

**Project administration:** Jill E. Weatherhead.

**Resources:** Jill E. Weatherhead.

**Software:** Alexander R. Kneubehl.

**Supervision:** Jill E. Weatherhead.

**Validation:** Yifan Wu, Nina L Tang, Alexander R. Kneubehl.

**Visualization:** Yifan Wu, Nina L Tang, Alexander R. Kneubehl.

**Writing – original draft:** Yifan Wu, Charlie Suarez-Reyes, Alexander R. Kneubehl, Jill E. Weatherhead.

**Writing – review & editing:** Yifan Wu, Nina L Tang, Alexander R. Kneubehl, Jill E. Weatherhead.

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
