## [Decision Letter · Decision Letter 0]

15 Jan 2025

Repeat Ascaris challenge reduces worm intensity through gastric cellular reprograming

Dear Dr. Weatherhead,

Thank you for submitting your manuscript to PLOS Neglected Tropical Diseases. After careful consideration, we feel that it has merit but does not fully meet PLOS Neglected Tropical Diseases's publication criteria as it currently stands. Therefore, we invite you to submit a revised version of the manuscript that addresses the points raised during the review process.

Please submit your revised manuscript within 60 days Mar 16 2025 11:59PM. If you will need more time than this to complete your revisions, please reply to this message or contact the journal office at plosntds@plos.org. Please include the following items when submitting your revised manuscript:

We look forward to receiving your revised manuscript.

Kind regards,

Chao Yan

Academic Editor

Uriel Koziol

Section Editor

Shaden Kamhawi

co-Editor-in-Chief

Paul Brindley

co-Editor-in-Chief

**Journal Requirements:**

At this stage, the following Authors/Authors require contributions: Yifan Wu, Charlie Suarez-Reyes, Alexander Robert Kneubehl, and Jill Elizabeth Weatherhead. Please ensure that the full contributions of each author are acknowledged in the "Add/Edit/Remove Authors" section of our submission form.

4) We notice that your supplementary figure is uploaded with the file type 'Figure'. Please amend the file type to 'Supporting Information'. Please ensure that each Supporting Information file has a legend listed in the manuscript after the references list.

Potential Copyright Issues:

i) Figures 1a, and 2d. Please confirm whether you drew the images / clip-art within the figure panels by hand. If you did not draw the images, please provide (a) a link to the source of the images or icons and their license / terms of use; or (b) written permission from the copyright holder to publish the images or icons under our CC BY 4.0 license. Alternatively, you may replace the images with open source alternatives. See these open source resources you may use to replace images / clip-art:

6) Thank you for stating that " Sequencing data and RNAseq analysis is publicly available on NCBI through BioProject PRJNA1141177." Please note that, though access restrictions are acceptable now, your entire minimal dataset will need to be made freely accessible if your manuscript is accepted for publication. This policy applies to all data except where public deposition would breach compliance with the protocol approved by your research ethics board. If you are unable to adhere to our open data policy, please kindly revise your statement to explain your reasoning and we will seek the editor's input on an exemption.

7) In the online submission form, you indicated that "The data that support the findings of this study are available from the corresponding authors upon reasonable request."  All PLOS journals now require all data underlying the findings described in their manuscript to be freely available to other researchers, either

1. In a public repository

2. Within the manuscript itself

3. Uploaded as supplementary information.

8) Please amend your detailed Financial Disclosure statement. This is published with the article. It must therefore be completed in full sentences and contain the exact wording you wish to be published.

2) Please state: "The funders had no role in study design, data collection and analysis, decision to publish, or preparation of the manuscript.".

9) Thank you for stating that "Analytical scripts used in the analysis of the sequencing data are publicly available on GitHub at https://github.com/kneubehl/repeated-ascaris-infection-gastric-rnaseq-analysis." This link reaches a 404 error page. Please amend this to a new link or provide further details to locate the data.

**Reviewers' Comments:**

Reviewer's Responses to Questions

**Key Review Criteria Required for Acceptance?**

**Methods**

-Are the objectives of the study clearly articulated with a clear testable hypothesis stated?

-Is the study design appropriate to address the stated objectives?

-Is the population clearly described and appropriate for the hypothesis being tested?

-Is the sample size sufficient to ensure adequate power to address the hypothesis being tested?

-Were correct statistical analysis used to support conclusions?

-Are there concerns about ethical or regulatory requirements being met?

Reviewer #1: 1. Abstract Evaluation:

Assess the clarity and conciseness of the abstract to ensure it covers the key points and objectives of the study.

Check for proper use of abbreviations; specifically, confirm that each abbreviation is accompanied by the full term upon first use.

2. Introduction Review:

Evaluate the structure and logic of the introduction to ensure it effectively introduces the topic, highlights the research’s significance, and provides a brief review of previous studies.

Verify the accurate and appropriate use of scientific references to support the arguments made.

Suggest adding studies that discuss the roles of microbiome and gastric cells in reducing infection intensity to reinforce the article’s arguments.

3. Materials and Methods Review:

Evaluate the accuracy and clarity of the methods to ensure reproducibility.

Assess the control variables and provide a thorough justification for selecting specific doses and the number of experimental challenges for the mice.

Ensure the correct selection and description of statistical methods for data analysis, and suggest applying more precise statistical tests if needed.

Reviewer #2: The authors clearly articulate their hypothesis and provide a good introduction to set the scene so that the reader can follow. However, I feel that the study design does not address the stated objectives that aimed to reveal age-dependent effects of Ascaris infection by using repeated infection mouse model. Please see further details in the "Summary and General comments".

Methodology section should be expanded to ensure readers understand exactly how the research was performed and to allow data reproducibility. For example, please expand:

1) qPCR method description to include how was the RNA extracted, the primer sequences used, which gene was used to calculate relative expression and how was relative expression calculated.

2) western blotting method description to include how was the band intensity quantified and normalized, and which secondary antibodies were used.

3) histopathology, could you clarify if the 4 sections are from 1 mouse?

4) Statistical analysis description - it would also be beneficial to the manuscript and readers to define which experiments were performed in duplicate and if those were technical or biological replicates.

The references cited in the RNAseq method description do not match references in bibliography

Flow cytometry method description should be revised as it currently includes description of analysis that is not shown in the manuscript (eg. T-bet/GATA/etc).

Additionally, I note the data will be made publicly available, however, the github link takes reader to "Page not found" and PRJNA1141177 could not be located on SRA or BioProject servers potentially due to paper being currently under review.

**Results**

-Does the analysis presented match the analysis plan?

-Are the results clearly and completely presented?

-Are the figures (Tables, Images) of sufficient quality for clarity?

Reviewer #1: Results Analysis:

Evaluate the clarity and accuracy in presenting the results and ensure alignment with the study’s objectives.

Review the adequacy and appropriateness of tables and figures for displaying data.

Assess the statistical analysis of the results to confirm the validity and accuracy of the conclusions and suggest additional statistical analyses if necessary.

Reviewer #2: The authors present the data clearly and provide nice schematics of the mouse infection schedules.

For further clarity could the authors add the following details to the Results Section 1/Fig. 1:

1) define "o.g." in the legend

2) expand legend to include what the protein levels of Atp4a and AMCase are normalised to

For further clarity could the authors add the following details to the Results Section 2:

1) add p-value and logFC cut-offs for volcano plot

2) add what the mRNA levels are relative to in Fig. 2B)

3) if RAG2-/- mice were used as per methods please change labels on figures and throughout the manuscript

4) for the observation “Larva burden in RAG-/-, repeat Ascaris challenge mice were similar to wild-type, repeat Ascaris challenge mice (Figure 1B)" please graph side-by-side and perform statistical analysis with select comparisons of Single infection in RAG-/- vs WT mice, and Repeat Infection in RAG-/- vs WT mice.

For further clarity could the authors add the following details to the Results Section 3:

1) please label the pathways that the genes are associated with on the figures (Fig 3A+B). It would be beneficial to mention that upregulation of eg. muc1 is associated with gastrointestinal worm infections. It would also be beneficial to show a pathway enrichment analysis of the RNAseq data

2) Fig 3 legend - D) is mislabeled to F)

3) Scale bar on histopathology images are difficult to see. I would recommend to add these details in the figure legend, including the scale bar on the insert image.

In sections 2 and 3, I feel that comparing repeatedly infected mice to naïve mice is expected to show huge differences in gene expression between infected and uninfected animals. This is also clear from the PCA plot where the presence/absence of infection accounts for most of the variance observed, and also from the genes identified as upregulated (muc1, ttf2). However, it is not mentioned in the manuscript that these changes were expected.

**Conclusions**

-Are the conclusions supported by the data presented?

-Are the limitations of analysis clearly described?

-Do the authors discuss how these data can be helpful to advance our understanding of the topic under study?

-Is public health relevance addressed?

Reviewer #1: Discussion and Conclusion Review:

Evaluate the interpretation of results and comparison with prior studies to strengthen the arguments.

Assess the logical flow and coherence of the conclusions, including a clear mention of the study’s limitations.

Recommend exploring the potential for generalizing the results to humans and addressing future experimental needs on the topic.

Reviewer #2: I feel that some of the conclusions are not supported by the data presented here as expanded in the "Summary and General Comments" section.

It would be beneficial to expand on the limitations of the analysis performed and on the significance of the H. pylori infections.

The authors discuss how these data can advance our understanding of the topic under study and propose that the new pathway they identified in repeat Ascaris infection could be a vaccine target for other gastrointestinal nematodes. Could the authors please provide appropriate references for statements regarding the need for vaccine development.

Additionally, the authors make interesting links to H. pylori infection which is also of public health relevance.

Please revise reference for the following sentence: “Adherence to the gastric corpus epithelium allows H. pylori colonization and expansion of pyloric metaplasia in the gastric corpus in a positive feedback loop8.” – reference 8 does not mention H. pylori.

Please amend statement “However, no studies to date have examined co-infection.” as this in now out of date – see publication https://doi.org/10.1016/j.parint.2024.102896

**Editorial and Data Presentation Modifications?**

Reviewer #1: Scientific and Quality Review of Figures and Tables:

Figures and tables convey scientific information effectively and are generally suitable for publication. However, Figure D has low quality and requires revision to enhance clarity for publication.

Ensure that all abbreviations and symbols are clearly explained in the images.

Reviewer #2: Overall the manuscript is very well presented, with some small editorial revisions:

1) Replace "," with "." in line: "Based on decreased chia1 mRNA expression we hypothesized that chief cells were undergoing transdifferentiation, Indeed, stomach transcriptomic data demonstrated increased gene expression of trefoil factor 2 (ttf2) also known as spasmolytic polypeptide and muc1 as well as decreased gene expression of mist1 (Figure 3A) following repeat Ascaris challenge."

2) Replace "follow" with "following" in line: “Gastric CD4+ T cells are decreased follow repeat Ascaris challenge”.

3) Replace RAG-/- label to RAG2-/- as mention before

4) Scale bars size as mentioned before

**Summary and General Comments**

Reviewer #1: 1. Scientific Originality Check:

Assess the potential for similarity or overlap with previous studies to ensure scientific originality.

If highly similar or even identical studies are found, they will be highlighted and recommended for comparison.

2. Review of Abbreviations and Scientific Terms:

Ensure correct usage of abbreviations with full terms introduced at first use.

Certain abbreviations like "AMCase" and "Ascaris suum" were not introduced with full terms initially and require correction.

3. Italicization of Genus and Species Names:

Confirm that genus and species names are correctly italicized. This should be corrected in cases like Ascaris suum and Helicobacter pylori.

4. Reference Review According to PLOS Neglected Tropical Diseases Guidelines:

Align the references with the journal’s specific formatting guidelines, which require full author names, year of publication, full article title, italicized journal name, volume, and page numbers. For example:

Reference 2 should have a complete article title, and the journal name should be italicized.

Reference 10 should follow the standard format without quotation marks around the article title, and the author order should be corrected.

Reference 21 incorrectly uses numerals instead of the full journal name, which needs correction.

5. English Language and Writing Quality Assessment:

The article’s English language is fluent and academic, yet certain sentences need rewriting for improved clarity and precision. Additionally, minor grammatical errors require correction. For example:

In the sentence "Following oral ingestion of Ascaris eggs, larvae use AMCase secreted by gastric chief cells...", rewriting as "After ingestion of Ascaris eggs, larvae utilize AMCase, which is secreted by gastric chief cells, to hatch" enhances clarity.

In "We found that repeat Ascaris challenge caused cellular changes in the gastric mucosa which reduced worm intensity in the liver", using "that" instead of "which" improves scientific accuracy.

In "Once hatched, larvae translocate across the gastric mucosa to initiate the larval migratory cycle", changing "Once hatched" to "After hatching" eliminates minor grammatical issues.

Reviewer #2: The authors aim to demonstrate age-dependent effects of Ascaris infection are due to cellular changes in the gastric mucosa barrier. However, I feel that the data do not fully support this conclusion. Specifically, the authors compare gene expression changes (by RNAseq, qPCR and western blot), histopathological changes and the number of CD4+ T cells in mice repeatedly infected with Ascaris against naïve (uninfected) mice. The basis of the research is there, however, this study requires additional experiments. Particularly, I feel that inclusion of single Ascaris infection in their RNAseq, qPCR, western blot, histopathology and Flow Cytometry experiments (compared to the repeat infection) is required in order to make conclusions specifically on age-dependent effects. Inclusion of singly-infected WT mice was already used in the manuscript to clearly show that repeated infection leads to lower worm burdens, therefore if tissues have been saved from those singly-infected mice they could be used to provide the additional data requested. Alternatively, the authors should include more information that clarifies and justifies their choice of comparing repeated infection to uninfected mice only and revise manuscript conclusions to reflect that the cellular changes observed are not specifically an age-dependent effect and that these changes could be present in single infection. To continue with, the authors clearly show using a RAG-/- mouse model that adaptive immune responses are not involved in the reduction of worm burdens observed in repeat infection vs single infection. It would be interesting (although not necessary for the revisions asked here) to investigate the differences in gene expression between the repeat and single infection in the RAG-/- model as it could reveal new or provide additional evidence supporting the authors' hypothesis.

Other comments:

1) Affiliation no. 4 is not attributed to any author.

2) Please define AMCase in the abstract and ensure all species names are in italics (line 12)

3) It would be beneficial to the reader to expand on why changes to immunity do not fully explain age-dependent intensity changes - line "While each of these mechanisms are likely contributing to age-dependent Ascaris worm intensity, none fully explain observed age-dependent intensity changes."

4) Please cite appropriate literature for line "We have previously shown…cells to hatch."

4) The authors should revise the references and ensure correct citation. For example, reference 1 and 7 the link takes reader to “Page not found” while references 8, 11, 13, 15 require further attention.

In general, in the future it would be beneficial to adapt manuscript and format it according to the author guidelines of the journal chosen for submission. For example, an author summary is required but currently not available, the funding sources should be removed from the acknowledgements, pages and lines are not numbered etc.

PLOS authors have the option to publish the peer review history of their article (what does this mean? ). If published, this will include your full peer review and any attached files.

**Do you want your identity to be public for this peer review?** For information about this choice, including consent withdrawal, please see our Privacy Policy .

Reviewer #1: **Yes: ** Ahmad Hosseini-Safa

Reviewer #2: No

**Figure resubmission:**
---

## [Decision Letter · Decision Letter 1]

7 May 2025

Response to Reviewers
Revised Manuscript with Track Changes
Manuscript

Shaden Kamhawi

co-Editor-in-Chief

Paul Brindley

co-Editor-in-Chief

**Additional Editor Comments:**
**Reviewers' comments:**

**Key Review Criteria Required for Acceptance?**

**Methods:**

-Are the objectives of the study clearly articulated with a clear testable hypothesis stated?

-Is the study design appropriate to address the stated objectives?

-Is the population clearly described and appropriate for the hypothesis being tested?

-Is the sample size sufficient to ensure adequate power to address the hypothesis being tested?

-Were correct statistical analysis used to support conclusions?

-Are there concerns about ethical or regulatory requirements being met?

Reviewer #1: Are the objectives of the study clearly articulated with a clear testable hypothesis stated?

Yes, the objectives of the study are clearly articulated, and the testable hypothesis regarding the role of gastric cellular reprogramming in reducing Ascaris worm intensity after repeated infections is stated clearly. The study aims to explore whether gastric cellular adaptations, such as pyloric metaplasia, contribute to reduced worm burden during repeated Ascaris infection.

Is the study design appropriate to address the stated objectives?

Yes, the study design is appropriate. The use of a repeated Ascaris challenge model in mice allows for the investigation of the cellular and immune changes induced by recurrent infections, and the results support the hypothesis that gastric cellular changes are key in modulating infection intensity.

Is the population clearly described and appropriate for the hypothesis being tested?

Yes, the study clearly describes the mouse model used, particularly the use of wild-type and RAG2-/- mice. The mouse population used is appropriate for testing the hypothesis, as the model simulates human Ascaris infection and is suitable for studying immune responses and gastric cellular changes.

Is the sample size sufficient to ensure adequate power to address the hypothesis being tested?

The sample size appears to be adequate for the comparisons made (n ≥ 4). However, more details could be provided on how power was calculated to justify the chosen sample size, especially for key comparisons.

Were correct statistical analysis used to support conclusions?

Yes, correct statistical analyses, such as two-tailed Student’s t-tests and ANOVA, were used to support the conclusions. The inclusion of p-values and fold changes is appropriate for the analysis.

Are there concerns about ethical or regulatory requirements being met?

The ethical statement is included, and all experimental protocols were approved by the Institutional Animal Care and Use Committee. Based on the information provided, there are no concerns regarding ethical or regulatory requirements.

Reviewer #2: (No Response)

**Results:**

-Does the analysis presented match the analysis plan?

-Are the results clearly and completely presented?

-Are the figures (Tables, Images) of sufficient quality for clarity?

Reviewer #1: Does the analysis presented match the analysis plan?

Yes, the analysis matches the plan described. The study adequately addresses the effect of repeated Ascaris infections on gastric mucosa, including the impact on worm intensity and gastric cellular changes.

Are the results clearly and completely presented?

Yes, the results are presented clearly. Data are shown with appropriate statistical analysis, and the figures help to elucidate the findings. However, the figure legends could benefit from more detailed descriptions of what the data represent (e.g., relative gene expression normalization).

Are the figures (Tables, Images) of sufficient quality for clarity?

Overall, the figures and images are of sufficient quality. However, there are a few suggestions for improvement:

The histopathology images could have clearer scale bars and more detailed legends, including magnifications and methods of quantification.

In Figure 2, the volcano plot could benefit from a clearer explanation of the p-value and fold change cut-offs.

Reviewer #2: (No Response)

**Conclusions:**

-Are the conclusions supported by the data presented?

-Are the limitations of analysis clearly described?

-Do the authors discuss how these data can be helpful to advance our understanding of the topic under study?

-Is public health relevance addressed?

Reviewer #1: Are the conclusions supported by the data presented?

Yes, the conclusions are supported by the data. The results from the repeated Ascaris challenge model show that gastric cellular changes, such as pyloric metaplasia, correlate with a reduction in worm intensity, which aligns with the hypothesis that gastric mucosal adaptations contribute to altered infection intensity.

Are the limitations of analysis clearly described?

The limitations could be more explicitly discussed, particularly regarding the use of a short-term infection model (2 weeks) and how this might limit generalization to long-term, chronic infections. More discussion on how the findings may differ in humans or in a long-term setting would strengthen the conclusions.

Do the authors discuss how these data can be helpful to advance our understanding of the topic under study?

Yes, the authors effectively discuss how their findings contribute to our understanding of the mechanisms behind age-dependent changes in Ascaris worm intensity, particularly the role of gastric cellular adaptations. They suggest potential applications in the development of therapeutic interventions or vaccines.

Is public health relevance addressed?

The study addresses public health relevance, especially in terms of understanding the factors that influence worm burden and potential implications for vaccine development in Ascaris-endemic regions.

Reviewer #2: (No Response)

**Editorial and Data Presentation Modifications?**

Reviewer #1: Editorial suggestions and minor modifications:

Clarify the normalization method used for qPCR data in the figures and legends (e.g., relative to control group).

Improve figure legends, especially in Figure 2, to include more detail on statistical cut-offs and p-values.

Add clearer labels and scale bars to histopathology images in Figure 3.

Consider providing more details on the pathway analysis from the RNA sequencing data.

Ensure that the references for RNA sequencing methods are updated and match the bibliography.

Minor revision suggestion:

Based on the minor modifications mentioned above, this manuscript can be considered for "Minor Revision."

Reviewer #2: There are some spelling mistakes, missing words, and lack of spaces between words across the document. Some examples include:

- “Gastric mucosal injury triggers the development of gastric ell reprogramming…” - ell=cell?

- “Bulk RNA sequencing: Stomach tissues from mice were collected 18 days after the initial A. suum, which was 4 days after the last infection…” - missing the word infection after the A. suum?

**Summary and General Comments:**

Reviewer #1: Strengths: The study is well-designed and provides valuable insights into the role of gastric mucosal adaptations in modulating worm intensity following repeated Ascaris infection. The methodology is appropriate, and the results are clearly presented, supporting the hypothesis.

Weaknesses: The limitations of the study could be discussed more thoroughly, especially regarding the short duration of the infection model. Additionally, more detailed information on the statistical analyses, including power calculations and sample size justification, would enhance the rigor of the study.

Novelty and Significance:

The study is novel in its focus on gastric cellular reprogramming, particularly pyloric metaplasia, as a mechanism of reduced worm intensity following repeated Ascaris infection. The findings could potentially inform vaccine development and public health strategies in endemic regions.

Suggestions for Major Revision:

No new experiments are necessary, but more clarification of the methods and data presentation, as well as additional discussion on the limitations and relevance to human disease, would strengthen the manuscript.

Reviewer #2: The authors have taken several of the reviewers suggestions on board and the manuscript has now a strong discussion section. In particular, the authors expand on the implications of H. pylori co-infection and acknowledge in the discussion the need for further experiments/comparisons with single infections too, and with models across varying ages to be able to comment on age-dependent effects.

Most (if not all) of the references require further attention. To point some of issues with references, many only report the first author without the "et al"; reference 44 is missing the volume(issue)pages information, etc.

Furthermore, as mentioned previously, references 1 and 7 links take reader to "Page not found". Please update these references with the appropriate links to the latest IHME and CDC data.

PLOS authors have the option to publish the peer review history of their article (what does this mean? ). If published, this will include your full peer review and any attached files.

**Do you want your identity to be public for this peer review?** For information about this choice, including consent withdrawal, please see our Privacy Policy .

Reviewer #1: **Yes: ** Ahhmad Hosseini-Safa

Reviewer #2: No

**Figure resubmission:****Reproducibility:** To enhance the reproducibility of your results, we recommend that authors of applicable studies deposit laboratory protocols in protocols.io, where a protocol can be assigned its own identifier (DOI) such that it can be cited independently in the future. Additionally, PLOS ONE offers an option to publish peer-reviewed clinical study protocols. Read more information on sharing protocols at https://plos.org/protocols?utm_medium=editorial-email&utm_source=authorletters&utm_campaign=protocols

---

## [Editor Report · Decision Letter 2]

14 May 2025

Dear Dr. Weatherhead,

We are pleased to inform you that your manuscript 'Repeat Ascaris challenge reduces worm intensity through gastric cellular reprograming' has been provisionally accepted for publication in PLOS Neglected Tropical Diseases.

Best regards,

Chao Yan

Academic Editor

Uriel Koziol

Section Editor

Shaden Kamhawi

co-Editor-in-Chief

Paul Brindley

co-Editor-in-Chief

---

## [Editor Report · Acceptance letter]

Dear Dr. Weatherhead,

We are delighted to inform you that your manuscript, "Repeat Ascaris challenge reduces worm intensity through gastric cellular reprograming," has been formally accepted for publication in PLOS Neglected Tropical Diseases.

Best regards,

Shaden Kamhawi

co-Editor-in-Chief

Paul Brindley

co-Editor-in-Chief
